# High Accuracy and Safety of Intraoperative CT-Guided Navigation for Transpedicular Screw Placement in Revision Spinal Surgery

**DOI:** 10.3390/jcm11195853

**Published:** 2022-10-02

**Authors:** Yen-Yao Li, Shih-Hao Chen, Kuo-Chin Huang, Chien-Yin Lee, Chin-Chang Cheng, Ching-Yu Lee, Meng-Huang Wu, Tsung-Jen Huang

**Affiliations:** 1Department of Orthopaedic Surgery, Chang Gung Memorial Hospital at Chiayi, Chiayi County 613016, Taiwan; 2Department of Medicine, College of Medicine, Chang Gung University, Taoyuan City 333323, Taiwan; 3Department of Orthopedics, Buddhist Tzu-chi General Hospital at Dahlin, Chiayi County 622401, Taiwan; 4Department of Nursing, College of Nursing, Chang Gung University of Science and Technology, Taoyuan City 333324, Taiwan; 5Department of Orthopedics, St Martin de Porres Hospital, Chiayi City 600044, Taiwan; 6Department of Orthopedics, Taipei Medical University Hospital, Taipei City 110301, Taiwan; 7Department of Orthopedics, School of Medicine, College of Medicine, Taipei Medical University, Taipei City 110301, Taiwan

**Keywords:** thoracolumbar spine, revision spinal surgery, intraoperative computed tomography, navigation, transpedicular screw

## Abstract

Background: Intraoperative CT-guided navigation (iCT-navigation) has been reported to improve the accuracy and safety of transpedicular screw placement in primary spinal surgery. However, due to a disrupted bony anatomy and scarring tissue, revision spinal surgery can be challenging. The purpose of this study was to evaluate the accuracy and safety of iCT-navigation for screw placement at the virgin site versus the revision site in revision thoracolumbar spinal surgery. Method: In total, 254 screws were inserted in 27 revision surgeries, in which 114 (44.9%) screws were inserted at the site with previous laminectomy or posterolateral fusion (the revision site), 64 (25.2%) were inserted at the virgin site, and 76 (29.9%) were inserted to replace the pre-existing screws. CT scans were conducted for each patient after all screws were inserted to intraoperatively confirm the screw accuracy. Results: In total, 248 (97.6%) screws were considered accepted. The rate of accepted screws at the virgin site was 98.4% (63/64) versus 95.6% (109/114) at the revision site (*p*: 0.422). There were six (2.4%) unaccepted screws, which were immediately revised during the same operation. There was no neurological injury noted in our patients. Conclusion: With the use of iCT-navigation, the rate of accepted screws at the revision site was found to be comparable to that at the virgin site. We concluded that iCT-navigation could achieve high accuracy and safety for transpedicular screw placement in revision spinal surgery and allow for the immediate revision of unaccepted screws.

## 1. Introduction

The transpedicular screw (TPS) system is one of the most widely used fixation devices in spinal surgery. It provides a three-column fixation [1] of the spine. It is essential to ensure the optimal trajectory of TPS insertion to offer the strongest bony purchase and prevent nerve tissue injury. TPS insertion can be challenging in revision spinal surgery [2], because the anatomical features used to identify the entry of a TPS may have been destroyed or obscured by the fusion mass or abundant scar tissue, causing a standard fluoroscopy to be insufficient to identify the proper entry of a TPS due to blurry images for altered or fused anatomy [2,3,4].

Several advanced image-guided techniques have emerged in the past two decades, including three-dimensional (3D) fluoroscopy [5,6], O-arm [7,8] and intraoperative computed tomography (iCT) [9,10]. One study [11] reported that the accuracy of the TPS was comparable for both primary and revision spine surgery using O-arm-guided navigation. However, some of the TPS placements in revision spinal surgery were performed at the virgin site for the extension of posterior instrumentation, instead of right over the revision site. Thus, the accuracy of TPS placement at the revision site could be different from that of the whole revision spine surgery. In the current study, we aimed to evaluate the accuracy and safety of iCT-navigation for TPS placement right over the revision site versus the virgin site in complex revision thoracolumbar spinal surgery.

## 2. Materials and Methods

We enrolled 27 consecutive patients who underwent TPS placement assisted with intraoperative CT (Siemens, Munich, Germany) integrated with navigation (Brainlab, Munich, Germany) for revision thoracolumbar spinal surgeries. The study was approved by the Institutional Review Board of the Chang Gung Memorial Hospital.

There were 27 patients (13 women and 14 men) in the present study, with an average age of 67 years (range 47–92 years). A total of 254 TPSs was inserted during the revision spinal surgeries, which were categorized into 3 groups: “revision site” screws (no. = 114) inserted over a previous laminectomy or posterolateral fusion field; “virgin site” screws (64) inserted at a site away from previous laminectomy or posterolateral fusion for the extension of posterior instrumentation; and “exchanged” screws (76), inserted to replace pre-existing screws with screws of a different style or brand through the previous trajectories.

### 2.1. Surgical Technique

After general anesthesia, the patient was positioned prone on a Wilson radiolucent frame (Mizuho OSI, California, CA, USA) on a Jackson table. A midline incision with paraspinal muscle stripping was performed to expose the previous operation site and extended to the virgin level if necessary. The pre-existing screws were exchanged for screws of a different brand, if necessary, which were inserted manually through previous trajectories without using iCT-navigation. Thereafter, the superior articular processes of the facets through which we planned to insert new TPSs were exposed. The reference array was then securely fixed at a vertebra distal to the expected lowest instrumented vertebra (usually L5 or S1) by tightly clamping its spinal process.

The operative field was scanned with the intraoperative CT, and the image data were transferred to the navigation system. A straight drilling guide was then registered to guide a 2.7 mm in diameter drill bit to create the optimal track for the TPS under iCT-navigation by checking the axial, sagittal and coronal views (Figure 1). A TPS with an optimal length and diameter was chosen and inserted through the pedicle trajectory after being adequately tapped. After completing all TPS placements, CT scans were conducted to confirm whether the TPSs were in the optimal position, which we called confirmatory CT scans (Figure 2).

### 2.2. Assessment of TPS Placement

Intraoperatively, with the aid of the confirmatory CT scan, we evaluated the TPS accuracy and divided the results into 2 conditions: “accepted” and “unaccepted”. An accepted screw was defined as having a medial pedicle breach of ≤3 mm or a lateral pedicle or body wall breach of ≤3 mm. The position of the TPS was considered biomechanically optimal and safe from neurologic injury. An unaccepted screw was defined as having a medial pedicle breach of >3 mm or a lateral pedicle or body wall breach of >3 mm (Figure 3); the TPS could be biomechanically suboptimal and create a risk of nerve tissue injury, and, therefore, was immediately revised under iCT-navigation. After replacing the unaccepted screw, the second confirmatory CT scan was performed to ensure that the screw was in an accepted position.

### 2.3. Radiographic and Clinical Analyses

The accuracy of the TPS placement was assessed with the first confirmatory CT. The rates of accepted screws and unaccepted screws were calculated for comparison between the virgin site and the revision site. Perioperative medical records were collected retrospectively, and postoperative neurological complications were evaluated.

### 2.4. Statistical Analysis

Descriptive statistics were reported as percentages to describe the rates of accepted screws and unaccepted screws. Fisher’s exact test was used to compare the rates of accepted and unaccepted screws between the virgin site and the revision site. The test was considered significant if *p* < 0.05.

## 3. Results

In total, 27 patients were categorized using the etiology for revision spine surgery: 12 patients (44.4%) were diagnosed with postlaminectomy instability with or without posterolateral fusion; 9 (33.3%) patients had adjacent segmental disease with spinal stenosis or instability; 4 (14.8%) patients had malpositions of a previous TPS with clinical neurologic symptoms; and the remaining 2 (7.4%) patients had a refractory postoperative infection (Appendix A).

Of the total 254 TPSs, 248 screw placements were considered accepted (97.6%) by the first confirmatory CT, including all of the “exchanged screws” (N = 76). The rates of accepted screws were comparable both at the virgin site (63, 98.4%) and the revision site (109, 95.6%). There was no significant difference between both groups (*p* = 0.422).

There were six screws (2.4%) considered unaccepted: one (1.6%) at the virgin site with a medial pedicle breach; and five (4.4%) at the revision site, including two with a medial pedicle breach and three with a lateral pedicle or body wall breach. The rates of unaccepted screws trended higher at the revision site than at the virgin site, but were not significantly different. All the unaccepted screws were revised immediately with the assistance of iCT-navigation, and, then, each proved to be accepted with a secondary confirmatory CT. No postoperative neurologic deterioration was observed related to the screw placement.

## 4. Discussion

With the use of iCT-navigation for revision spinal surgery, the rate of accepted screws was 97.6%, which was comparable to other studies [2,11] for revision surgery. Hsieh et al. [11] adopted an O-arm with navigation to guide TPS placement. They reported “good or fair” rates of screw placement (pedicle breach <3 mm) as 98.7% in the primary surgery group and 98.6% in the revision surgery group, which were not significantly different. Even compared with our previous studies for primary surgery [10,12], which reported the accuracy of TPS placement was 96–98%, the accuracy in the current study for revision surgery was comparable. However, we divided our series of TPS placements into two groups for comparison: the revision site and the virgin site. The rate of accepted screws was 98.4% (63/64) at the virgin site and 95.6% (109/114) at the revision site (*p* = 0.422). There was one (1.6%) unaccepted screw noted at the virgin site, and five (4.4%) at the revision site. The rates of unaccepted screws trended higher at the revision site than at the virgin site, but were not significantly different. We assumed that some of the TPSs in revision spinal surgery could be inserted at the virgin site for the extension of posterior instrumentation instead of right over the revision site. Thus, the accuracy of TPS placement at the revision site could be different from or inferior to that at the virgin site across the whole revision spinal surgery.

In the current study, most of the unaccepted screws were noted at the revision site. We considered dense postoperative fibrotic tissue or scarring to be the principal causes resulting in screw malpositioning at the revision site. Once the navigation device bent around the dense scar during the process of TPS placement, the navigation had the potential to fail. The scar tissue caused excessive pressure on the instrument and usually resulted in the medial deviation of the screw trajectory. Furthermore, if the surgeon exerted excessive bending force on the instrument (the drill or awl) to apply soft tissue pressure, the instrument could skive off the pedicle and result in the lateral skidding of the trajectory. The skiving effect might explain why there were more unaccepted screws presenting with a lateral wall breach (3, 60%) at the revision site. We suggest that the skiving effect could be diminished with the adequate release of soft tissue tension, and by using a high-speed drill with a shard bit for drilling [13].

There have been few studies focused on the feasibility or accuracy of TPS placement in revision spinal surgery. Kim et al. [2] reported that the accuracy of TPS placement in revision surgery was excellent with the free-hand technique. The revision rates of misplaced screws were 1.1% in the virgin site screw group and 2.9% in the revision site group, which were not significantly different. However, the TPSs were evaluated with plain radiography, not CT, which might have underestimated the number of misplaced screws. Free-hand TPS insertions in revision spinal surgery are technically demanding, and it is laborious to identify the entry point of the TPS when anatomic features are distorted. With the use of iCT-navigation in the current study, surgeons were able to confidently insert the TPSs during revision spinal surgery. Hsieh et al. [11] adopted an O-arm (Medtronic, Inc., Minneapolis, MN, USA) with navigation to guide TPS placement. They collected postoperative CT scans to evaluate screw accuracy in revision spinal surgery. Interestingly, for the purpose of decreasing radiation, confirmatory intraoperative CT scans were not routinely performed in their study, unless there was concern for screw malpositioning. There were six poor screws (1.4%, pedicle breach of > 3 mm) found postoperatively in their study; fortunately, none caused neurological deficits and, thus, were not revised. In our study, unaccepted screws were identified intraoperatively with the first confirmatory CT. After the secondary confirmatory CT, all the revised screws were accepted.

Since unaccepted screws may not always result in neurologic deficits, it might not be necessary to revise all of them intraoperatively. Laine et al. [14] reported that no screw caused neurological problems if the pedicle breach was less than 4.0 mm on the postoperative CT scan. Castro et al. [15] found that a TPS with medial pedicle breach of more than 6 mm created a high risk of nerve root damage. In the current study, the criteria to revise a screw was a medial pedicle breach of > 3mm or a lateral pedicle or body wall of >3 mm, which might be appropriate. However, we should keep in mind that even a medial 2 mm breach can be symptomatic, whereas a lateral breach of approximately 5 mm often remains asymptomatic in some extreme case. Although all unaccepted screws were revised intraoperatively and successfully, the use of a secondary confirmatory CT increased the patients’ exposure to radiation. Intraoperative neuromonitoring was available in the current study, by which the authors could have avoided revising some of the unaccepted screws and rescued a TPS with a pedicle breach of <3 mm and abnormal neuromonitoring alerts.

In spinal surgery with posterior instrumentation, the free-hand technique combined with intraoperative fluoroscopy is commonly used to confirm the entry point and trajectory of the TPS. The reported incidence of misplaced screws ranged from 10.5% to 21% [14,16,17,18]. Two-dimensional (2D) fluoroscopy may not provide sufficient image information to safely insert TPSs in spinal deformities [12,17,18] due to a remodeled bony structure or spinal fractures [10,16] because of hypermobile spinal segments and disrupted anatomy, such as facets or pedicle fractures. Furthermore, it would be technically challenging to use fluoroscopy for TPS placement in revision spine surgery due to the altered anatomy, indistinct bony landmarks and excessive scar tissue [2]. CT scans can provide adequate image information to evaluate TPS position, including the axial, coronal and sagittal planes [14], and is, therefore, the better choice for guiding TPS placement in cases of compromised spine anatomy, such as deformity, trauma, and revision spine surgery.

In the last two decades, several advanced imaging techniques have emerged that are able to provide 3D images. Usually, 3D fluoroscopy is limited by a small “field-of-view” scan area and an inferior image quality, with a screw accuracy inferior to iCT [19]. The O-arm is a cone-beam computed tomography, which has higher rates for screws revised intraoperatively [20], but has a similar accuracy for postoperative screw placement as compared with iCT. Therefore, we believe that iCT is the imaging technology of choice to provide the optimal image resolution for TPS placement. However, it cannot provide real-time imaging to immediately determine the screw trajectory and may further expose the patients to a greater radiation dosage.

Certain navigation techniques enable the diminution of CT scans for guiding TPS placement, while providing virtual but real-time images to guide TPS placement. The iCT-navigation technique is an effective method for TPS placement in complex spinal surgery for patients with spinal deformities [12] and unstable thoracolumbar spine fractures, with a high TPS accuracy of up to 98% [9,10]. It also allows for the immediate revision of unaccepted TPSs during surgery, followed by a secondary confirmatory CT scan. This option is highly practical in the case of immediate intraoperative revision, because returning to another surgery for screw revision would increase the risk for the patient. Although the confirmatory CT scan exposes patients to radiation, the medical staff are protected from radiation in a lead-shielded room.

There were several limitations in our study. It might be relatively expensive to set up the iCT-navigation system; however, the installation charge should be considered against the cost of reoperation for patients with misplaced screws. We should have measured the dose of radiation to which patients were exposed, which might have been relatively high, since the entire workflow of iCT-navigation surgery contained at least two CT scans, including the registration scan and confirmatory scan. According to our previous studies [10] for primary surgery, the mean dose of patient radiation exposure was 15.8 mSv, and the mean dose per single level was 2.7 mSv. Fortunately, according to the recommendation by the International Commission on Radiological Protection, the annual maximum permissible dose is 20 mSv per year with no single year exceeding 50 mSv. Hence, the radiation dose to the patient in our study might have been within the limit. Furthermore, although this study was purely an imaging-based investigation and did not focus on functional outcomes, it was found that there were no immediate postoperative neurologic deficits.

## 5. Conclusions

The use of iCT-navigation for TPS placement in revision thoracolumbar spinal surgery was a useful technique for improving the accuracy and safety of TPS placement and allowed for the immediate intraoperative revision of unaccepted screws. The rate of accepted screws at the revision site was comparable to that at the virgin site in revision spinal surgery, although the rate of unaccepted screws at the revision site trended higher than at the virgin site. No postoperative neurological injury was noted in our patients.

## Figures and Tables

**Figure 1 jcm-11-05853-f001:**
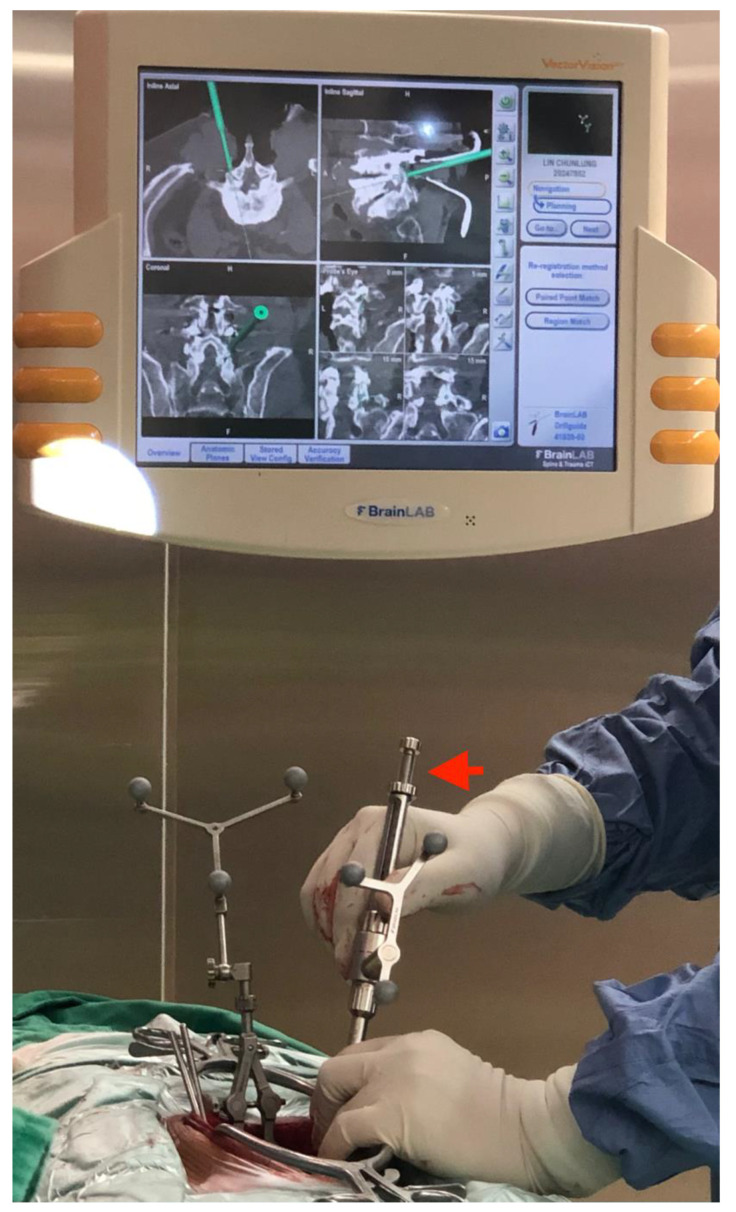
The straight drilling guide (red arrow) was used to guide a 2.7 mm in diameter drill bit to create the optimal TPS track under iCT-navigation by checking the axial, sagittal and coronal views.

**Figure 2 jcm-11-05853-f002:**
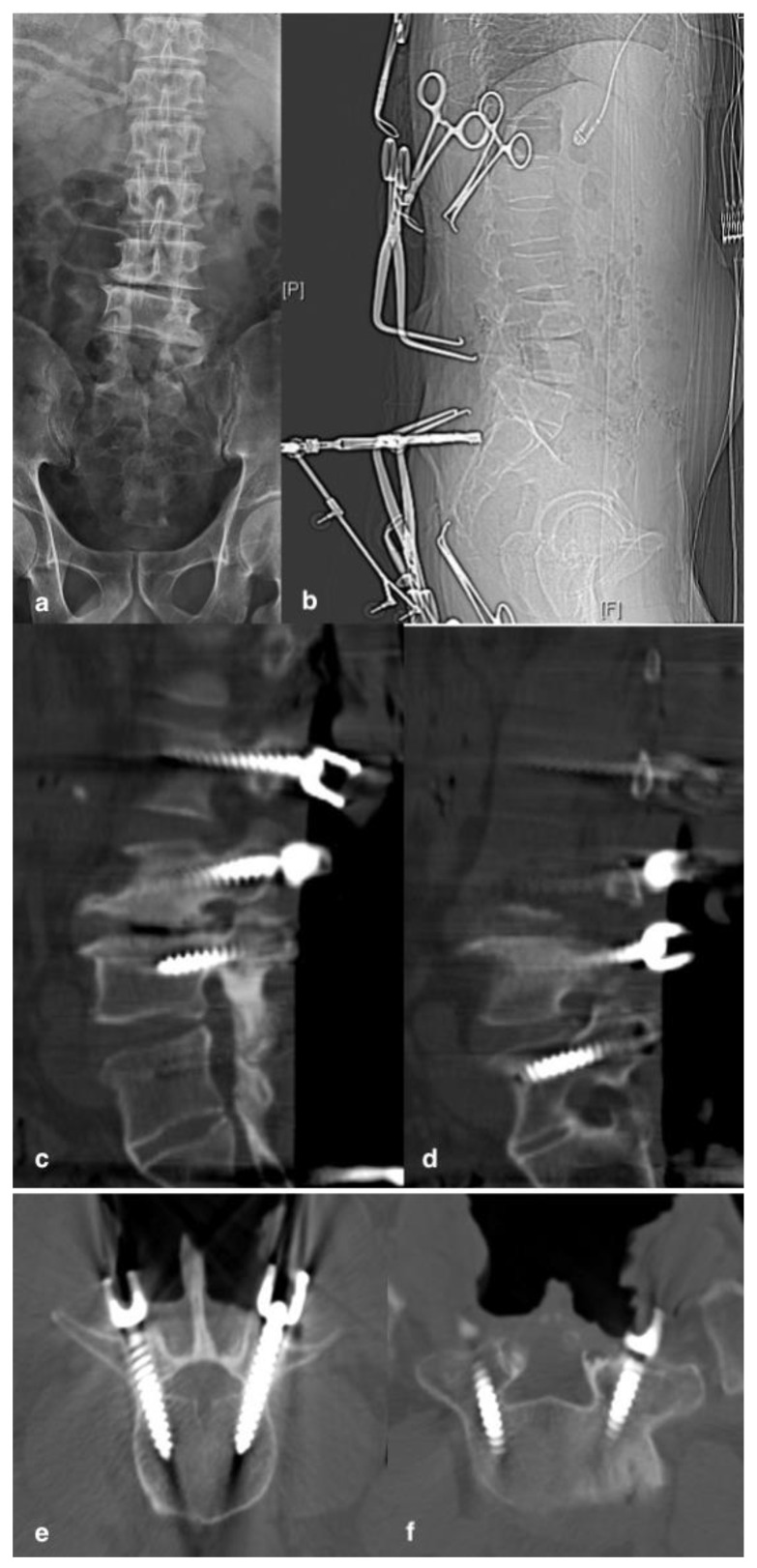
A 55-year-old male patient with post-laminectomy instability and spinal stenosis; X-ray (**a**) showed previous laminectomy of L4, L5 and partial L3. This patient underwent revision spinal surgery with the use of iCT-navigation; X-ray (**b**) showed the reference array fixed at the spinal process of S1 for iCT-navigation. Intraoperatively, after all screws were inserted, the confirmatory CT scan demonstrated sagittal and axial views of screws (**c**,**e**) at the virgin site and (**d**,**f**) at the revision site.

**Figure 3 jcm-11-05853-f003:**
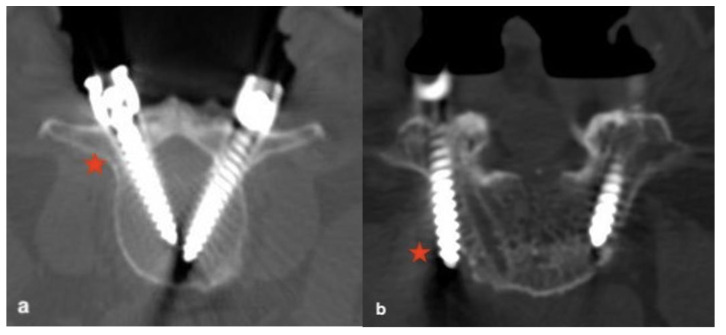
Unaccepted screw (red star) was defined as (**a**) a medial pedicle breach of >3 mm or (**b**) a lateral pedicle or body wall breach of >3 mm.

## Data Availability

The original data generated during the study are included in the article. Further inquiries can be directed to the corresponding author.

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
