# Peer review of "High Accuracy and Safety of Intraoperative CT-Guided Navigation for Transpedicular Screw Placement in Revision Spinal Surgery"

_jcm, 2022, doi:10.3390/jcm11195853_

Round 1
Reviewer 1 Report
The authors present an interesting, apparently prospective unrandomized, comparative study of intraoperative ct navigation use for confirming the placement of thoracolumbar screws. From the author's review of the literature, it would seem that their study presents novel information. Significant English language (grammar and style) will be required for this manuscript. Table 1 should be incorporated as a supplementary table and table 2 should be incorporated into the text.Author Response
Please see the attachment.

Reviewer 2 Report
One criticism of the presented work is that at least 2 CT scans of the patient were performed in the entire workflow. This represents a considerable radiation exposure for the patient (especially if preoperative CT scans were also taken). If this procedure is the standard workflow, it should be reconsidered in terms of patient protection.
Another point of criticism is that only revision patients were examined here, without making a comparison with patients who had undergone primary surgery from the authors' own cohort. It is of interest whether the accuracy of primary operated patients in the same setting is different.
The main criticism is the classification of pedicle screws as accepted and unaccepted. Exceeding the bone limit by more than 3 mm is assumed to be the benchmark here. This classification is inaccurate. A classification according to the degree of perforation, such as the 2 mm increment classification, would be beneficial. The authors should consider re-evaluating the imaging of the patients and classify according to one of the known classification systems. This would provide a more complete picture. It is known that only a few pedicle screw breaches become symptomatic. Even a medial 2 mm breach can be symptomatic, whereas a lateral breach around 5 mm often remains asymptomatic. Since the screws were revised intraoperatively in this case, I believe that accurate grading is critical to determine the extent of malpositioning.
Reviewer 3 Report
Actually, the authors’ purpose of current study, ‘to evaluate the accuracy and safety of the iCT-navigation for TPS placement at right over the revision site’, cannot be attained by this case-series with such limited number of cases having no controls. There have been numerous studies which investigated comparative clinical and radiologic outcomes of navigation techniques. The authors should deliberately consider changing the design of their study by including proper matched controls and asserting originality of their study precisely.
Author Response
Dear Sir/Madam,
Thank you very much for your valuable comments and concerns, which reminds me to clarify the novel design of the current study although there have been numerous studies which investigated comparative clinical and radiologic outcomes of navigation techniques. we also had reported two studies (ref. 10 and 18) which evaluated the accuracy of TPS placement in primary surgery for spinal trauma and adolescent idiopathic scoliosis with the iCT-navigation.
However, we believed that a revision spinal surgery usually contained screw placement right over the revision site and others at the virgin site, which actually mimics primary surgery. Therefore, we conducted this study since we believed that the accuracy of screw placement in revision spinal surgery might be overestimated since across the revision surgery there were some of TPSs placement in the virgin site mimicking the condition in primary surgery.
Thanks a lot again for your concerns and suggestions.
Thank you very much for your attention.
Sincerely,

Round 2
Reviewer 3 Report
Thank you for your reply regarding my previous comments. However, as an original study to assess the accuracy and safety of intraoperative navigation, I still think it is a prerequisite to select appropriate controls. Although you have reported the accuracy of TPS placement in primary surgery in other reports, those results cannot be used for comparison due to possible differences in factors associated with surgical outcome, such as different types of disease, different spinal regions or segments, different spinal instrument, surgical skill or technique, etc.
I recommend the authors to compare their results with those from their separate controls who underwent primary or revision surgery (using TPS at virgin sites) in their institution during the same period using conventional methods without navigation. The separate controls should be similar to the cases in terms of such factors.
Author Response
Dear Sir or Madam,
Thank you again for your comments. I appreciate the challenges from an expert. As your recommendation, by which we could certainly compare our results of TPS placement in revision surgery (using iCT-navigation) with those in primary surgery or revision surgery without iCT-navigation. Such the comparison is not the purpose of the current study because we would rather make a comparison on the results with iCT-navigation in between revision surgery and primary surgery.
However, we consider that a revision spinal surgery usually contains partial screw placements right over the revision site and others at the virgin site which actually mimics primary surgery. The accuracy of screw placement in revision spinal surgery might be overestimated because only TPSs placement in the revision site will meet the distorted anatomy due to previous surgery, but not in the virgin site. Hence, we design the method to make a comparison between the revision site and the virgin site (mimicking primary surgery).
As you said that it is a prerequisite to select appropriate controls, we completely agree with you. The variable factor in the current study had been controlled such as different types of disease, different spinal regions or segments, or different spinal instrument, because we actually compared the results in between revision site and the virgin site across the same surgery (or patient), thus the variation (bias) could be controlled to few or none.
We have already reviewed other studies comparing the results using navigation in revision surgery with those in primary surgery. However, We just doubted that the accuracy of TPS placement in revision surgery could have been overestimated since some screws in the virgin site could be inserted as the same (easy) as those in primary surgery. That is why we conducted the work with the different study design.
We appreciate you providing such challengeable and valuable comments.
Hope to get your understanding.
Sincerely
